# Multimodal joint deconvolution and integrative signature selection in proteomics
Yue Pan[1], Xusheng Wang [2,3], Jiao Sun[1], Chunyu Liu [4], Junmin Peng [5,6] & Qian Li [1] ✉

Deconvolution is an efficient approach for detecting cell-type-specific (cs) transcriptomic signals without cellular segmentation. However, this type of methods may require a reference profile from the same molecular source and tissue type. Here, we present a method to dissect bulk proteome by leveraging tissue-matched transcriptome and proteome without using a proteomics reference panel. Our method also selects the proteins contributing to the cellular heterogeneity shared between bulk transcriptome and proteome. The deconvoluted result enables downstream analyses such as cs-protein Quantitative Trait Loci (cspQTL) mapping. We benchmarked the performance of this multimodal deconvolution approach through CITE-seq pseudo bulk data, a simulation study, and the bulk multi-omics data from human brain normal tissues and breast cancer tumors, individually, showing robust and accurate cell abundance quantification across different datasets. This algorithm is implemented in a tool MICSQTL that also provides cspQTL and multi-omics integrative visualization, available at https://bioconductor.org/packages/MICSQTL.

Proteomics profiling and analysis at cell type level is critical in the study of complex biological systems with numerous applications in immunology, cancer research, and developmental biology[1–3]. Several technologies have been developed to identify and quantify proteins at cellular resolution[4]. For example, the detection of proteins by CyTOF coupled with fluorescence activated cell sorting (FACS)[5] and the single cell multimodal technology CITE-seq[6], which is a multimodal sequencing technique that enables simultaneous profiling of gene expression and up to 300 cell surface protein markers in individual cells, allows the identification of rare cell types and cells that express low levels of certain genes. However, these technologies only measure limited number of proteins. The abundance of proteins not detectable in CyTOF or CITE-seq may be strikingly different from the transcript expression of coding genes and cannot be approximated by scRNA-seq measurement because of the RNA/protein degradation and post-translation modifications.

Recent advances in liquid chromatography mass spectrometry (LC-MS)-based proteomics methods have addressed the limitations in the sensitivity and throughput[7,8], which accelerates the evolvement of single cell mass spectrometry (scMS) proteomics. One major challenge in scMS

proteomics is that the number of unique samples and cells analyzed in a single day is very limited[9]. For label-free scMS, samples are analyzed sequentially with analysis time ranging from 35 to 90 min. At this speed, a maximum of 40 single cells could be analyzed in a day, which is not ideal for population-scale clinical studies due to the burden of time and costs. The scMS technology limitations and costs of cell sorting are the hurdles for cell-type-specific inference such as differential expression (csDE) or protein quantitative trait mapping (cspQTL) that requires median or large sample size.

Deconvolution algorithms are rapidly developed to measure molecule proportions (e.g., RNA transcripts) mapped to each cell type, which is different from the cell counts composition and varies across molecular sources. To estimate the cellular composition in human proteome, the pure cell or single cell reference proteomes (i.e., signature matrix) is needed but lacking in certain tissues or cell types due to the challenges in cellular dissociation (e.g., astrocytes and excitatory neurons[2]) and the aforementioned limitations in scMS and CITE-seq. Meanwhile, the multi-omics profiling matched by samples becomes popular in recent decade, enabling the integration across data sources and the discovery of

[1]Department of Biostatistics, St. Jude Children's Research Hospital, Memphis, TN 38105, USA. [2]Center for Proteomics and Metabolomics, St. Jude Children's Research Hospital, Memphis, TN 38105, USA. [3]Department of Genetics, Genomics & Informatics, University of Tennessee Health Science Center, Memphis, TN 38105, USA. [4]Department of Psychiatry, SUNY Upstate Medical University, Syracuse, NY 13210, USA. [5]Department of Structural Biology, St. Jude Children's Research Hospital, Memphis, TN 38105, USA. [6]Department of Developmental Neurobiology, St. Jude Children's Research Hospital, Memphis, TN 38105, USA. ✉e-mail: qian.li@stjude.org

multimodal signatures for disease[1]. Hence, we design a algorithm to estimate the proteomics cell fractions by integrating bulk transcriptome-proteome without single cell reference proteome, implemented in R package MICSQTL. Our method enables the downstream cell-type-specific protein quantitative trait loci mapping (cspQTL) based on the mixed-cell proteomes and pre-estimated proteomics cellular composition, without the need for large-scale single cell sequencing[10] or cell sorting.

## Results

### MICSQTL algorithms and implementation

Our method quantifies the cell abundances in proteins by jointly deconvoluting matched bulk transcriptome and proteome, which can be used in downstream cspQTL mapping (Fig. 1). For each tissue sample with bulk transcriptome-proteome, we model the cellular compositions $\theta^{(i)}$ in the $i$th modality as a product of tissue-specific cell counts fractions $p$ and molecule source-specific cell size factors $s^{(i)}$. A Joint Non-negative Matrix Factorization (JNMF) framework was employed to link the modalities through shared cell counts $p$, allowing individualized multi-modal reference panels. We employ a loss function that integrates the observed bulk RNA and protein expressions to optimize the cell abundances in each molecular source, as described in Methods. The parameters in JNMF are initialized by an RNA signature matrix of similar tissue type and the RNA proportions pre-estimated by CIBERSORT (CBS)[11] with this signature matrix, the first of which can be obtained from scRNA-seq or sorted cell RNA-seq data accessible in many public repositories. Hence, this joint deconvolution algorithm is semi reference-free without using a single cell or pure cell proteomics reference profile, implemented in the function *deconv*.

In proteomics deconvolution, researchers may not have a priori knowledge about the cell marker proteins to be used for certain cellular subpopulations, but the cell marker genes in transcriptome have been broadly identified and curated in public databases[12]. Here, we use the AJIVE framework[13] to construct a common space shared across two molecular sources: bulk RNA expression of cell marker genes and the sample-matched whole proteome, which captures the between-sample heterogeneity caused by cellular abundance variation. Next, the observed whole proteome is projected onto this shared space by employing the reduced-rank loadings from AJIVE, where protein dimensions and annotations are unchanged. The rank of loadings is determined by an inherent algorithm. The potential cell marker proteins are selected based on the feature-wise Euclidean distance between the projected and observed proteomes. This cross-source feature selection is similar to ReFACTor[14], but ReFACTor is only applicable to single modality and sets the rank of loadings by the assumed number of cell types. Hence, we name our signature selection procedure as 'AJ-RF' and use the selected proteins in joint deconvolution, implemented in the function *ajive_decomp*.

### Validation using multimodal expression from CITE-seq

We first validate our algorithm by using the pseudo bulk multimodal expression profiles built from a public CITE-seq dataset with 161,764 human peripheral blood mononuclear cells (PBMCs). These samples were collected from eight donors[1] and processed by 10 × 3′ technology and Seurat v4. For each single cell, the expression of 228 surface proteins and more than 30,000 RNA genes were measured. Hence, we generated pseudo bulk expression data for a list of RNA cell marker genes and 228 surface proteins by aggregating feature-wise abundance or Unique Molecular Identifier (UMI) counts across the cells per donor. The ground truth of cellular fractions can be achieved by the annotated cell labels, which are identical for RNA and protein in CITE-seq.

We selected four donors (samples P1 and P7 in Fig. 2a) with disparities in B cell, natural killer (NK) cell, dendritic cell (DC), and other T cell abundances to generate the pseudo signature matrix. The cell count fractions quantified by our algorithm were similar to the true cell count fractions captured and annotated in CITE-seq (i.e., Pearson correlation (r) = 0.91, Lin's concordance correlation coefficient (CCC) = 0.91), as demonstrated in Fig. 2b, and showing improvement over the CBS method (r = 0.88, CCC = 0.85). The individualized cell-type-specific expression is exemplified for three cell types: B cell, CD4 T cell, and CD8 cell in Fig. 2c–h. Although the individualized cell-type-specific protein expression displayed larger variance, it remained significantly correlated with the observed pure cell bulk expression (Fig. 2c–e). This discrepancy may arise from the distinct nature of cell surface protein measurement through a binding strategy, as opposed to RNA UMI counts. The RNA expression levels resolved by our algorithm were well-aligned near the diagonal line with $r > 0.9$ across all three cellular populations (Fig. 2f–h). Further, we examined the impact of varying step sizes on the results obtained from the same input data and different rescaling approaches. The results in Supplementary Note 1 Figs. S1–S5 suggest that despite of ambiguity in the optimal step size, moderate adjustments in step size and rescaling by log or MinMax do not lead to substantial change in the deconvoluted cellular composition. Nevertheless, the above results demonstrate the validity and power of our algorithm without reliance on a single cell (type) proteomics reference.

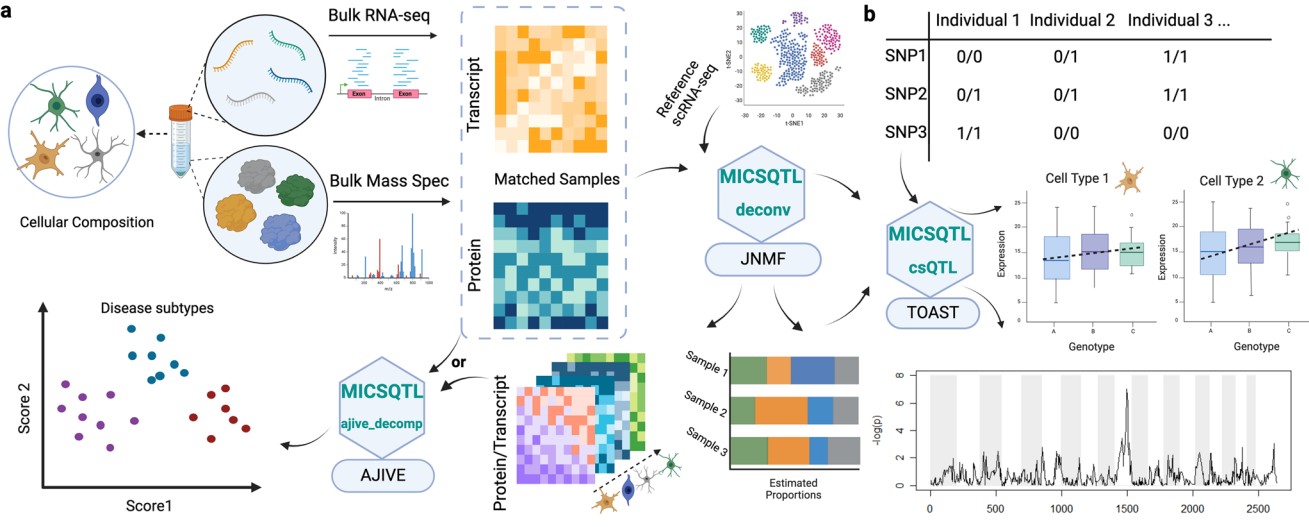

**Fig. 1 | Flowchart of MICSQTL. a** Cross-source joint deconvolution and integration of matched transcriptome-proteome. **b** Cell-type-specific pQTL based on estimated cell fractions. This figure was created by using BioRender.com.

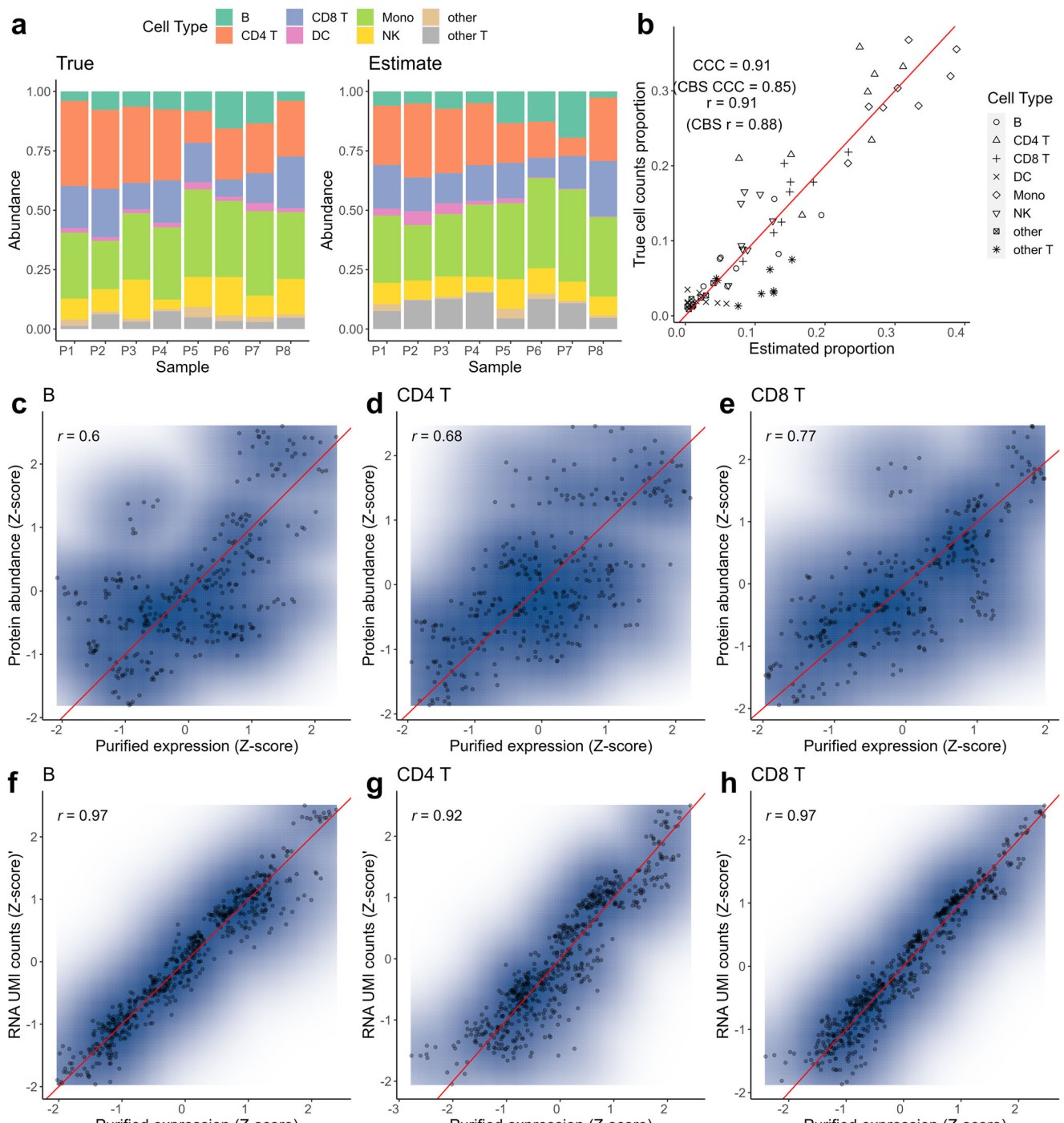

**Fig. 2 | The performance of JNMF in CITE-seq pseudo bulk data. a** Composition plots for true (left) and JNMF estimated (right) cell counts fractions. **b** A scatterplot for true vs. JNMF estimated cell counts fractions with Pearson correlation (r) and Lin's concordance correlation coefficients (CCC), whereas CBS metrics are in parentheses ($n$ = 8 biologically independent samples). **c–e** Scatterplots for the true abundance vs. digitally-purified expression of surface proteins per sample in B cells, CD4 and CD8 T cells. **f–h** Scatterplots for the true UMI counts vs. digitally-purified RNA expression per sample in B cells, CD4 and CD8 T cells. The source data can be found in Supplementary Data 1.

## Assessment using simulation data

The above pseudo bulk data from CITE-seq only provides the ground truth for cell counts fractions instead of the cell proportions in each molecule source. To mimic the possible differences in RNA vs. protein cellular compositions and compare the performance of distinct methods, we rigorously designed a simulation study to generate synthetic bulk transcriptome-proteome with ground truth of modality-specific cellular compositions. The statistical models used in data generation are described in

Supplementary Note 2, in which the statistical parameters are extracted from a public scRNA-seq data of human brain[15], a public single-cell-type mouse brain proteomics data[16], and a bulk proteomics data of human brain described in the next section. We designed two scenarios to allow relatively low (scenario A) and high (scenario B) correlations between the protein and RNA proportions, as visualized in Fig. 3a, b.

We compared our algorithm to three existing methods: the RNA proportions resolved by CBS (i.e., the initial cell counts fractions used in

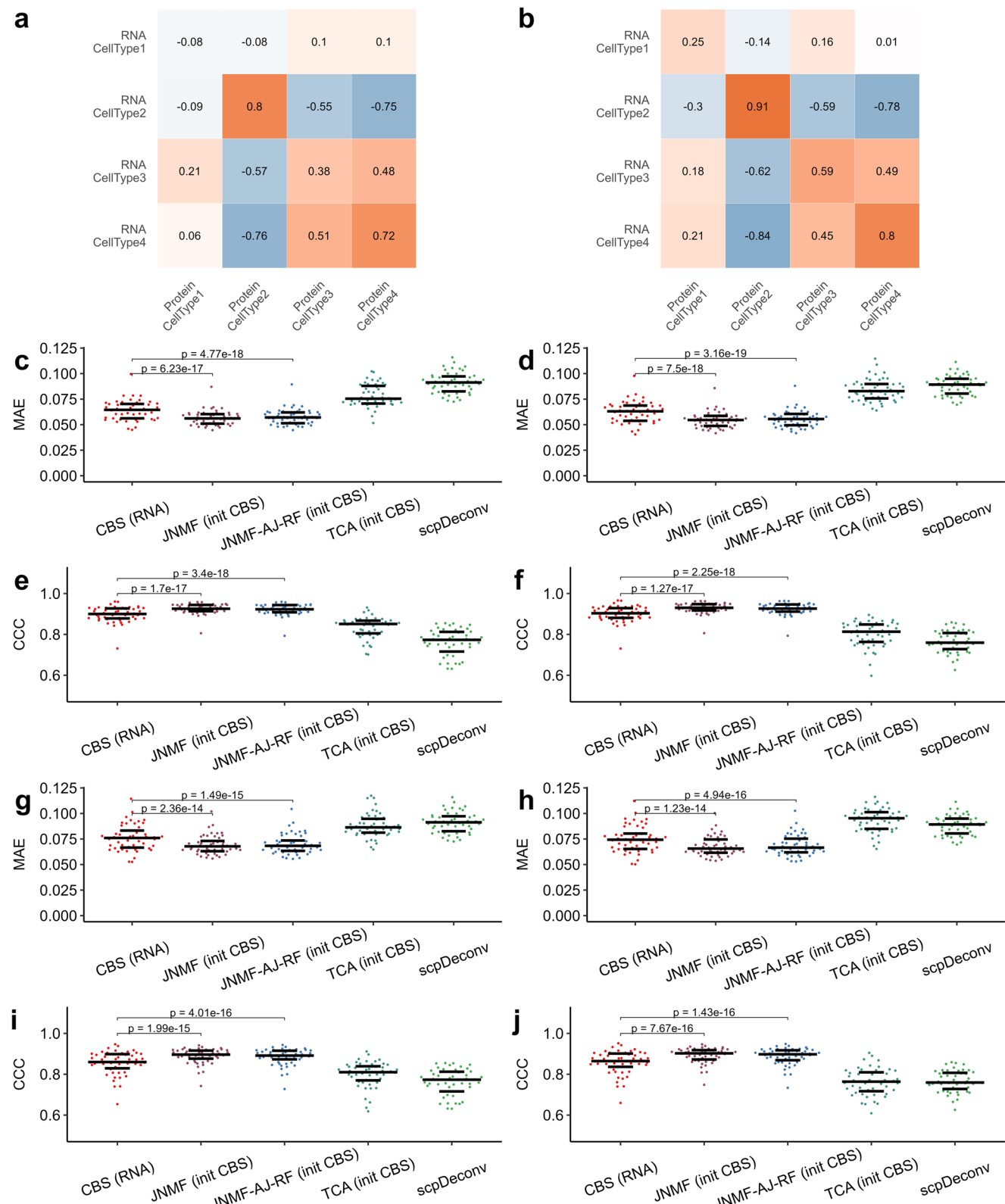

**Fig. 3 | The performance of competing deconvolution methods in simulation study. a, b** Pearson correlation coefficients between synthetic protein and transcript proportions per cell type in scenarios A and B, respectively (*n* = 100 biologically independent samples for each simulation set); (**c–f**) Mean absolute error (MAE) and CCC for each deconvolution method based on pseudo RNA signature matrices with small noises per scenario; (**g–j**) MAE and CCC for each method based on pseudo RNA signature matrices with large noises. The *p* value of paired t test on the initial vs. output proportions of JNMF are shown in panels (**c–j**). The source data can be found in Supplementary Data 2 (scenario A) and Supplementary Data 3 (scenario B).

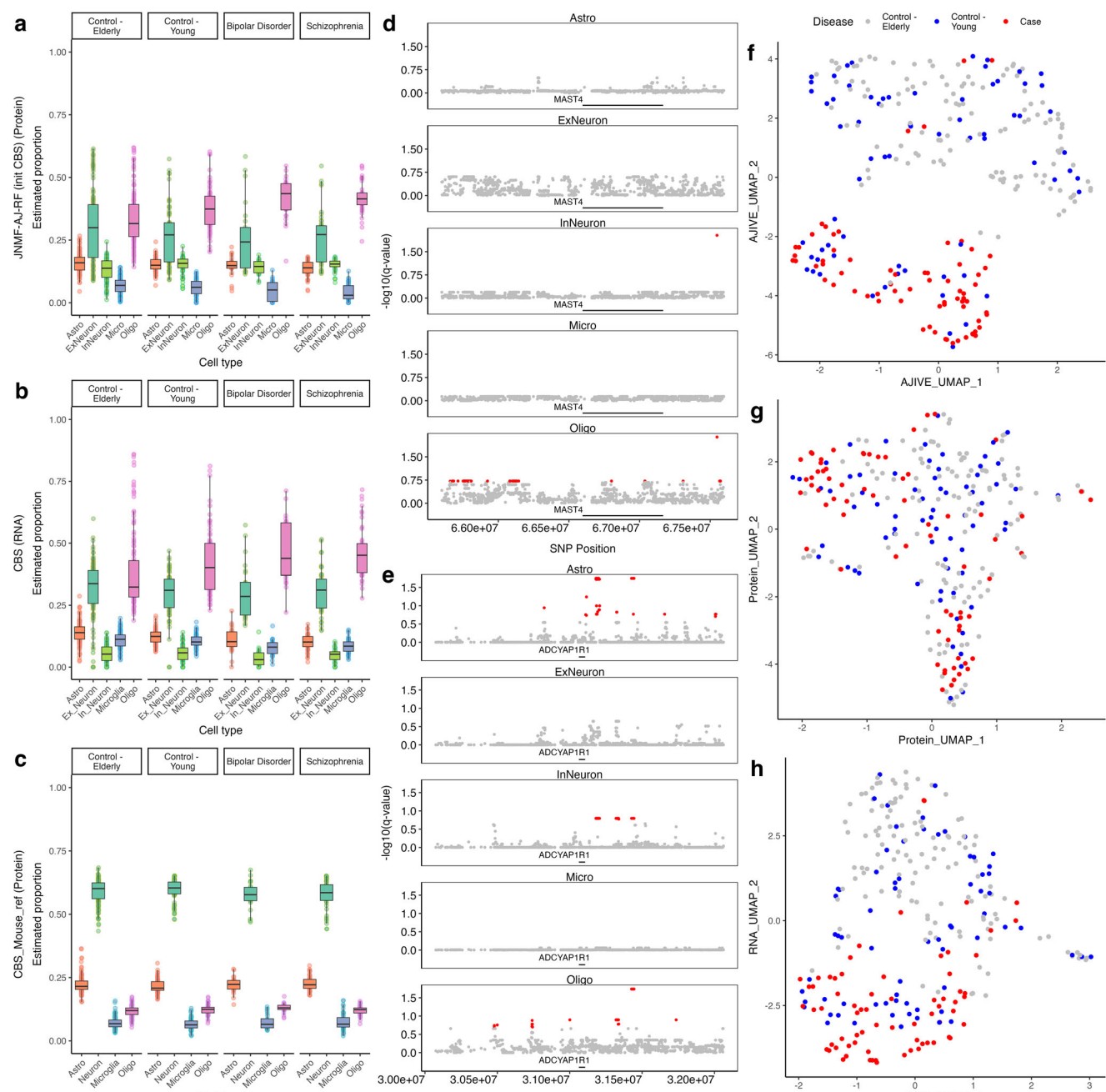

**Fig. 4 | An application of MICSQTL to the human prefrontal cortex multi-omics data. a** Protein proportions in human prefrontal cortex quantified by JNMF-AJ-RF with initial values as (**b**) RNA cell proportions quantified by CBS. **c** Protein proportions in human prefrontal cortex quantified by CBS with mouse brain pure cell reference ($n = 264$ biologically independent samples). **d, e** FDR-controlled adjusted $p$ value for cspQTL analysis of proteins *MAST*4 and *ADCYAP*1*R*1. **f–h** UMAP for joint and single-source visualizations. The source data can be found in Supplementary Data 4.

JNMF) as surrogate protein proportions, the cellular fractions estimated by TCA with CBS RNA proportions as initial value and the bulk proteomes as target data, and a deep learning-based method scpDeconv[17] using single-cell reference data for training. To the best of our knowledge, the high-quality single cell proteomics data in human prefrontal cortex is not publicly available. Therefore, we chose the scRNA-seq data used in the above data generation as a surrogate single cell reference for scpDeconv. The output from each method was compared to the ground truth protein proportion via mean absolute error (MAE) in Fig. 3c–d, g–h and CCC in Fig. 3e–f, i–j.

To deconvolve the synthetic bulk RNA-seq data with CBS, we constructed $n = 50$ replicates of pseudo signature matrix per scenario by introducing small vs. large random noises to the ground truth of subject-specific reference transcriptome. The RNA cell marker genes were chosen based on gene-wise coefficients of variation from the true reference panel, while 700 out of 1000 proteins were selected with AJ-RF. For each pseudo signature matrix and the corresponding initial CBS estimate, JNMF significantly improved the accuracy of cellular compositions in proteomes compared to CBS, as validated by a paired $t$ test (Fig. 3c–j). Our method also outperforms TCA and scpDeconv across the scenarios and pseudo signature matrices, whereas a single cell (type) proteomics reference profile is lacking. Notably, the proteins selected by AJ-RF yielded similar outperformance over the competing methods, implying the efficacy in capturing the latent cellular heterogeneity.

We also benchmarked the robustness of our algorithm via initialization with Non-Negative Least Squares (NNLS) RNA proportions and assessed

**Fig. 5 | Application to breast cancer tumor samples. a** Reference scRNA-seq cell counts fractions. **b** Result of CBS with signature matrix built form scRNA-seq data in (**a**). **c** Result of JNMF-AJ-RF with initial values as the signature matrix from (**a**) and RNA proportions from (**b**). **d** Result of scpDeconv with single cell CyTOF proteomics reference data (*n* = 122 biologically independent samples). The source data can be found in Supplementary Data 5.

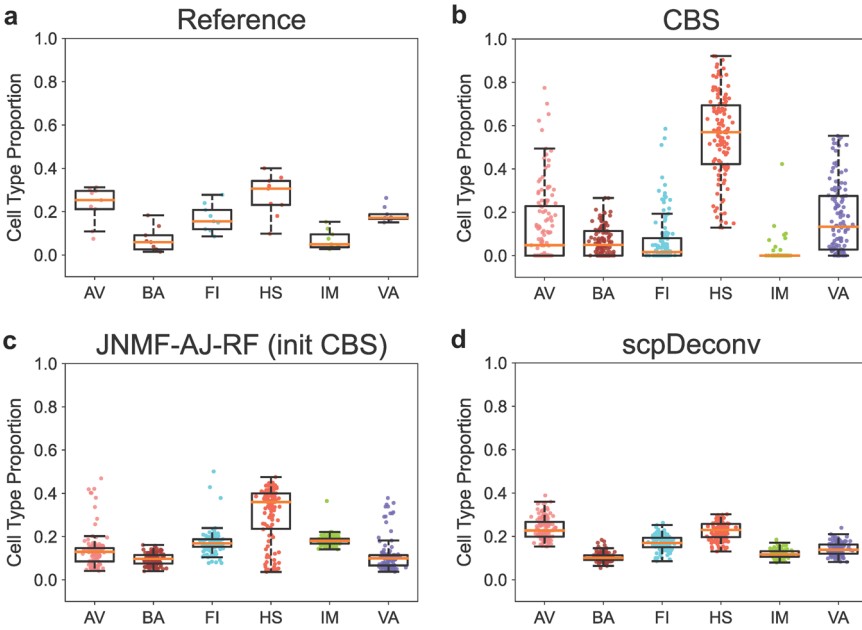

the computation power of scpDeconv in bulk RNA deconvolution. Supplementary Fig. S6 presents statistically significant improvement by JNMF in MAE and CCC, although NNLS estimate was less accurate compared to CBS initial and restricted the absolute performance of our method. The result of proteomics deconvolution tool scpDeconv in an application to bulk RNA-seq data was even worse than NNLS because of the inherent differences between RNA and protein molecules and the disitnct profiling technologies. Again, our algorithm demonstrated superior accuracy and robustness in predicting proteomics cell proportions compared to alternative approaches.

**Application to bulk transcriptome-proteome in bipolar disorder, schizophrenia, and healthy controls**

We used tissue-matched transcriptome-proteome from postmortem human brain prefrontal cortex in a study of (*n* = 25) bipolar disorder (BP) and (*n* = 45) schizophrenia (SCZ) cases and (*n* = 194) healthy controls to demonstrate the performance of MICSQTL. The details of human brain tissues, MS proteomics and RNA-seq transcriptomics profiling are available in Supplementary Note 3. The signature genes used in competing methods were selected from previous findings[16], while the (initial) signature matrix was generated by scRNA-seq data of healthy human prefrontal cortex[15]. The protein proportions quantified by JNMF-AJ-RF algorithm (Fig. 4a) showed possible disparities in astrocyte, microglia, and oligodendrocyte abundances between the (BP and/or SCZ) cases and controls whose brain tissues (death) were under 70 years of age. This result aligns with the expected cell abundance in human prefrontal cortex[15,18] and is similar to the previous findings about glia cell abundance in SCZ[19,20]. But there are confounding factors not regressed out in this dataset such as sub-cohorts and aging. The validity of cellular compositions in BP and SCZ compared to controls requires more experiments or an external cohort with adequate sample size of single cell proteomes or transcriptomes. On the other hand, the initial CBS RNA deconvolution failed to recover the expected mean abundances in inhibitory neuron and microglia (Fig. 4b). We also applied CBS to the bulk proteomes with a proteomics signature matrix of four cell types profiled from major regions of mouse brain[16], which neither decomposed the neuron cells into excitatory and inhibitory subpopulations nor recovered the expected cell abundances (Fig. 4c).

A downstream cspQTL screening was performed by cell-type-specific differential analysis, implemented in the function *csQTL*. The input data

include bulk proteomes, whole genome sequencing genetic variants (SNP), and protein proportions in Fig. 4a and Supplementary Data 4. We selected several mutation markers reported in previous literature to illustrate the csQTL function in MICSQTL. For each protein of interest, we select the nearby SNPs within a genomic distance of 1 million bases (Mb). Figure 4d, e shows the SNPs with false discovery rate (FDR) adjusted *p* value < 0.2 per cell type for the proteins encoded by genes associated with neurodevelopmental or neuropsychiatric illness: *MAST4* and *ADCYAP1R1*. These genes or the related gene family were reported as mutations associated with increased risks for Alzheimer's disease[21], mega-corpus-callosum syndrome[22], and post-traumatic stress disorder[23]. The adjusted p-values were listed in Supplementary Data 4, while cspQTL results for additional genes were shown in Figure S7. Note that the validity of cspQTL result depends on the study populations for the bulk proteomes and the accuracy of proteomics deconvolution, which should be investigated through a rigorous single cell study measuring the proteins of interest with adequate sample size. Last but not least, our tool MICSQTL outputs the multivariate Common Normalized Scores (CNS) from AJIVE that represents the sample-specific variation shared across transcriptome and proteome, which may uncover the heterogeneity across disease phenotypes and the hidden drivers. The multi-omics human brain tissue samples were jointly visualized in Fig. 4f by using the concatenated CNS, which outperforms the single modality visualization (Fig. 4g, h).

**Application to multi-omics data in breast cancer**

To demonstrate our method in different tissue types, we utilized the patient-matched bulk RNA-seq and MS proteomics data from *n* = 122 breast cancer (BC) tumor tissues[24] along with the single cell multi-omics data in an external BC cohort[25]. We first applied CBS to the bulk RNA-seq data[24] and a signature matrix built from the external scRNA-seq profiles[25], which served as the initial values in JNMF algorithm. The aforementioned deep learning deconvolution tool scpDeconv was applied to the bulk MS proteomics data, using the single cell CyTOF proteomics profiles from external BC tissues[25] to train the autoencoder. The input single cell and bulk data were transformed by log scale for CBS, JNMF-AJ-RF and z-score for scpDeconv.

The deconvolution result by each method was benchmarked by the annotated cell fractions in the scRNA-seq data of external BC tissues[25] (Fig. 5a). Obviously, the tumor microenvironment deconvolved by CBS with bulk transcriptomes (Fig. 5b) was substantially improved by applying

JNMF-AJ-RF to the bulk multi-omics profiles (Fig. 5c and Supplementary Data 5). The proteomics tumor microenvironment composition resolved by a scpDeconv submodule that only uses the (seven) proteins detected in both single cell CyTOF panel and bulk MS proteomes was the best estimate (Fig. 5d). However, this outperformance of scpDeconv may rely on the accurate measurement or low dimension of marker proteins in the CyTOF reference panel and not necessarily hold for the untargeted single cell reference proteomics data (e.g., scMS). To illustrate this caveat, we ran another submodule of scpDeconv that imputed the single cell reference for additional (20 or 50) highly variable proteins (HVP) only detected in the bulk proteome. For either set of HVP, this module was rerun ten times to assess the variation of performance. The replicates of deconvoluted proportions in Supplementary Note 1 Figs. S8–S9 demonstrate that predicting the abundance of proteins not available in CyTOF reference panel may randomly reduce the accuracy of deconvolution. In other words, the performance of scpDeconv depends on the availability and quality of protein markers measured at single cell level.

## Discussion

Our pipeline offers three primary functions to perform multi-omics cell abundance quantification with or without marker protein selection, integrative visualization, and cspQTL mapping. The semi reference-free joint quantification of cellular compositions in RNA and proteins were benchmarked by multiple datasets. That is pseudo bulk RNA and protein expression constructed from CITE-seq single cell data, synthetic bulk multimodal expression generated by statistical models, and real human brain bulk transcriptome-proteome with external snRNA-seq data.

Overall, JNMF coupled with cross-modality signature protein selection significantly improves the cell abundance quantification of MS proteomes compared to CBS, TCA, and scpDeconv with imputation. Our algorithm also identifies the proteins contributing to the latent cellular heterogeneity in bulk multi-omics profiles, which may elucidate the cell marker proteins for certain tissue types. This multimodal deconvolution framework is more favorable in population-scale studies compared to the single modal deconvolution or single cell profiling, since it neither relies on a single cell proteomics reference profile nor requires cell clustering and labeling.

The hyperparameter (PGD step size) in our algorithm has marginal impact on the estimated proportions and can be adjusted according to the scale of input data. In the current version of MICSQTL, we do not suggest the optimal scaling approach since the accuracy of purified data varies and may affect the downstream analysis such as integrating purified multi-omics profiles. According to the extensive experiments in CITE-seq pseudo bulk data, the impact of step size on cellular fractions was reduced by the MinMax rescaling across all the features, but the purified RNA expression was distorted.

Another potential utility of our tool is the high-resolution purification of individualized pure cell expression, which is an essential component in the output of our algorithm and paves the way for deep proteome profiling at single cell type resolution. However, the current implementation of joint deconvolution algorithm emphasizes the accuracy of cell abundance quantification, which may sacrifice the power of high-resolution purification. A possible solution to improve individualized multimodal pure cell expression is to train a deep learning model (such as autoencoder[26]) on the observed bulk proteomes and the reference panels pre-estimated by JNMF algorithm. Meanwhile, it's worth employing Stochastic Gradient Descent in the future to reduce the errors in pre-estimated reference panels.

Altogether, the JNMF deconvolution algorithm substantially improves the cell abundance estimation in bulk proteome by integrating modalities and using PGD for high-dimensional parameter optimization. The impact of PGD optimization on the quantified cell abundances and individualized purification will be assessed extensively in future. Our tool MICSQTL not only fills the methodological gap in bulk proteomics deconvolution without using single cell

proteomics data, but also sheds light on the design of a comprehensive experiment that profiles single cell MS proteomes matched to bulk samples to benchmark the performance of different deconvolution tools.

## Methods

### Multimodal joint deconvolution

For the tissue biospecimen of individual $i$, we measure the expressions of protein $g$ ($g = 1, ..., G$) and mRNA transcript (or gene) $m$ ($m = 1, ..., M$), respectively, denoted by $y_{gi}^{(1)}, y_{mi}^{(2)}$. The unobserved and individualized pure cell expressions are denoted by $x_{gik}^{(1)}, x_{mik}^{(2)}$. The molecular source-specific cellular fractions for cell type $k$ are $\theta_{ik}^{(1)}, \theta_{ik}^{(2)}$ ($k = 1, ..., K$), determined by the common tissue-specific cell counts (fractions) $p_{ik}$ and source-specific cell size factors $s_{ik}^{(1)}, s_{ik}^{(2)}$. That is $\theta_{ik}^{(1)} = p_{ik}s_{ik}^{(1)}$, $\theta_{ik}^{(2)} = p_{ik}s_{ik}^{(2)}$. Thus, the bulk multi-modal expression data are modeled as

$$E(y_{gi}^{(1)}) = \sum_{k=1}^{K} x_{gik}^{(1)}\theta_{ik}^{(1)} = \sum_{k=1}^{K} x_{gik}^{(1)}p_{ik}s_{ik}^{(1)},$$

$$E(y_{mi}^{(2)}) = \sum_{k=1}^{K} x_{mik}^{(2)}\theta_{ik}^{(2)} = \sum_{k=1}^{K} x_{mik}^{(2)}p_{ik}s_{ik}^{(2)}.$$

We denote $\boldsymbol{y}_i^{(1)} = \left[y_{gi}^{(1)}\right]_{G \times 1}$, $\boldsymbol{X}_i^{(1)} = \left[x_{gik}^{(1)}\right]_{G \times K}$, $\boldsymbol{y}_i^{(2)} = \left[y_{mi}^{(2)}\right]_{M \times 1}$, $\boldsymbol{X}_i^{(2)} = \left[x_{mik}^{(2)}\right]_{M \times K}$, $\boldsymbol{p}_i = \mathrm{diag}(p_{i1}, ..., p_{iK})$, $\boldsymbol{s}_i^{(1)} = \left[s_{ik}^{(1)}\right]_{K \times 1}$ and $\boldsymbol{s}_i^{(2)} = \left[s_{ik}^{(2)}\right]_{K \times 1}$. The above source-specific models are linked by the common tissue-specific cell counts fractions $\boldsymbol{p}_i$. Hence, we propose to jointly estimate the high-dimensional non-negative parameters $\boldsymbol{\eta}_i = \{\boldsymbol{X}_i^{(1)}, \boldsymbol{X}_i^{(2)}, \boldsymbol{p}_i, \boldsymbol{s}_i^{(1)}, \boldsymbol{s}_i^{(2)}\}$ by minimizing a loss function that integrates $\boldsymbol{y}_i^{(1)}$, $\boldsymbol{y}_i^{(2)}$. This is achieved by solving

$$\hat{\boldsymbol{\eta}}_i = \arg\min_{\boldsymbol{\eta}_i}\left\{ \left\|\boldsymbol{y}_i^{(1)} - \boldsymbol{X}_i^{(1)}\boldsymbol{p}_i\boldsymbol{s}_i^{(1)}\right\|^2 + \left\|\boldsymbol{y}_i^{(2)} - \boldsymbol{X}_i^{(2)}\boldsymbol{p}_i\boldsymbol{s}_i^{(2)}\right\|^2 \right\}$$

subject to $\min\{\boldsymbol{\eta}_i\} \geq 0$.

This loss function integrates the observed multimodal bulk data by the shared cell counts $\boldsymbol{p}_i$ as an extension to the Non-negative Matrix Factorization. This algorithm initializes the multi-omics reference panels $\boldsymbol{X}_i^{(1)}, \boldsymbol{X}_i^{(2)}$ and the cell counts fractions $\boldsymbol{p}_i$ with an external RNA-seq signature matrix and the RNA proportions pre-estimated by CBS. These sample-wise parameters of multimodal reference panels and cellular compositions are then optimized by the Projected Gradient Descent algorithm[27], being simultaneously updated and adapted to tissue-specific bulk multi-omics profiles.

### Algorithm.

1. The initial values of $\boldsymbol{X}_i^{(1)}, \boldsymbol{X}_i^{(2)}$ are the cell-type-specific expression from an external RNA signature matrix, while the initial values of $s_{ik}^{(1)}, s_{ik}^{(2)}$ are ones. The genes in initial $\boldsymbol{X}_i^{(2)}$ should be mapped to the features in bulk proteome $\boldsymbol{y}_i^{(2)}$. Initialize $\boldsymbol{p}_i$ by applying CIBERSORT to the target bulk RNA-seq data with the above (RNA) signature matrix.
2. Evaluate the loss function's first gradient at current estimates: $\boldsymbol{l}(\boldsymbol{\eta}_i^{(s)})$.
3. Update parameters with non-negative bounds: $\boldsymbol{\eta}_i^{(s+1)} = [\boldsymbol{\eta}_i^{(s)} - \Delta\boldsymbol{l}(\boldsymbol{\eta}_i^{(s)})]_+$, where $\Delta$ is step size.
4. Repeat steps 2-3 until convergence: $\max\{|\boldsymbol{\eta}^{(s+1)} - \boldsymbol{\eta}^{(s)}|\} < \epsilon$ or a limit of iterations (e.g., 1000).
5. Normalize the cellular fractions as $\theta_{ik}^{(1)} = \frac{p_{ik}s_{ik}^{(1)}}{\sum_{k=1}^{K} p_{ik}s_{ik}^{(1)}}$ and $\theta_{ik}^{(2)} = \frac{p_{ik}s_{ik}^{(2)}}{\sum_{k=1}^{K} p_{ik}s_{ik}^{(2)}}$.

## Integrative signature selection

The decomposition provided by AJIVE enables the identification of underlying biological patterns that are common to all molecular modalities. Suppose there are $J$ molecule sources for the same sets of $N$ samples but different features from each source. For the bulk expression matrices $Y^{(j)}$ ($j = 1, \ldots, J$) in distinct modalities, AJIVE integrates $Y^{(j)}$'s to reconstruct each by three components: $Y^{(j)} = C^{(j)} + I^{(j)} + E^{(j)}$, where $C^{(j)}$ represents the common variation originating from the $j$th modality, $I^{(j)}$ and $E^{(j)}$ are the source-specific structured variation and the residual noise, respectively.

The top proteins contributing to the common variation shared between proteome ($Y^{(1)}$) and signature genes ($Y^{(2)}$) are selected by using the loadings $V^{(1)}$ obtained from the singular value decomposition (SVD) of matrix $C^{(1)} = V^{(1)} D^{(1)} U^{(1)}$, where $V^{(1)}$ is $G \times r$ with $r$ as reduced rank estimated by the Wedin bound procedure[13]. Next, we compute $\tilde{Y}^{(1)} = V^{(1)} V^{(1)^T} Y^{(1)}$ as a projected approximation to the observed bulk proteome with rank $r$. For each protein $g$, we calculate the distance between $y_{g\cdot}^{(1)}$ and $\tilde{y}_{g\cdot}^{(1)}$ by the Euclidean distance $d_g = \| y_{g\cdot}^{(1)} - \tilde{y}_{g\cdot}^{(1)} \|$. To select the proteins that contribute most significantly to the shared variation, we choose the proteins with the smallest distances $d_g$.

## Cell-type-specific protein QTL mapping

The potential errors produced in sample-wise deconvoluted proteomes may lead to bias in cell-type-specific QTL mapping. Hence, we apply a published method implemented in TOAST[28] to perform cell-type-specific protein differential analysis for genotypes based on the bulk proteomes, estimated protein cell proportions, and whole genome sequencing variants. This method uses pre-estimated cell proportions and a linear model to describe the cell-type-specific differential expression pattern in bulk data, and then performs F-test on the hypothesized cell-type-specific changes across three genotype groups[29], with false discovery rate being well controlled according to the previous simulation studies[30,31].

## Reporting summary

Further information on research design is available in the Nature Portfolio Reporting Summary linked to this article.

## Data availability

The processed signature matrices derived from both real datasets are available for access on the GitHub repository: https://github.com/YuePan027/MICSQTL/tree/main/processed_signature. The CITE-seq data is available on GEO under accession number GSE164378. The human brain prefrontal cortex mass spectrometry proteomics data and RNA-seq data are available in the Synapse database https://www.synapse.org under accession code syn32136022. The breast cancer bulk multi-omics data are available as supplementary files in[25], while the scRNA-seq and single cell CyTOF data of breast cancer are available at GEO: GSE180878 and https://data.mendeley.com/datasets/vs8m5gkyfn/1. The source data necessary for creating the main figures can be found in Supplementary Data 1-5.

## Code availability

MICSQTL is a Bioconductor package with license GPL-3, available at https://bioconductor.org/packages/MICSQTL. CIBERSORT (https://cibersortx.stanford.edu/). TOAST (https://bioconductor.org/packages/TOAST). TCA (https://cran.r-project.org/web/packages/TCA).

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

## Acknowledgements

This work was partially supported by Cancer Center Support Grant P30CA21765 (Y.P., Q.L.), the American Lebanese Syrian Associated Charities (Y.P., X.W., J.S., J.P., Q.L.), and NIH R01MH110920 grant (C.L.).

## Author contributions

Y.P., X.W., J.P., and Q.L. conceive this study. Y.P. develops the algorithms and R package as maintainer, performs simulation and real data analyses, and visualizes analysis results. Q.L. proposes the algorithm and designs R package, simulation study, and real data analysis. J.S. runs experiments on scpDeconv in simulation study and breast cancer data, and visualizes the results. Q.L. and Y.P. write the manuscript. X.W., C.L., and J.P. generate and share the real data, contribute to methodology discussion, and interpret real data analysis results. All authors reviewed the manuscript.

## Competing interests

The authors declare no competing interests.
