## [Peer Review File · Communications Biology]

Reviewers' comments:

Reviewer #1 (Remarks to the Author):

The Pan et al introduces a novel algorithm, MICSQTL, designed for dissecting bulk proteomes by leveraging information shared between transcriptome and proteome data. The method identifies potential cell marker proteins and quantifies cell abundance in mixture proteomes, enabling downstream analyses such as cell-type-specific protein Quantitative Trait Loci (cspQTL) mapping. The authors validate the algorithm using various datasets, including pseudo bulk expression profiles from CITE-seq, synthetic bulk multimodal expression, and real human brain transcriptome-proteome. The results demonstrate the validity and power of MICSQTL, particularly in accurately estimating cell abundances in proteins. The tool is applied to study bipolar disorder and schizophrenia, showcasing its potential to reveal disparities in astrocyte and oligodendrocyte abundances between cases and controls. The tool yields more accurate cell proportion estimates in the prefrontal cortex compared to Cibersort (referred to as CBS), effectively representing the actual distribution, such as accurately depicting the prevalence of oligodendrocytes and maintaining a 1:3 ratio between Inhibitory (In) and Excitatory (Ex) neurons. Nevertheless, I ask some questions about these results later - as well as for cspQTL result that was also used to demonstrate the utility of the tool.

Questions & Comments:

[reported differences between scz-bd-control] The authors report disparities in astrocyte and oligodendrocyte abundances between cases and controls. However, there is a limited discussion about the direction of that difference (higher in controls or vice versa?) and how it matches the existing findings in terms of the directionality of the change. It is not clear whether the input matrix was somehow normalized / residualized, i.e. whether the effect of technical covariates was regressed out. Are those reported differences statistically significant? Would it be possible to generate sample-level estimates as well as per-cell-type exact outputs (p-values & adj. p-values) as supplementary tables? Is there any other cohort that could be used to replicate these findings? I am asking also before I previously saw papers reporting the changes mostly (but not only) in glutamatergic neurons.

[cspQTL] The results should be better described quantitatively (e.g. as Supplementary Table) as well as in the biological context (are those implicated genes established in SCZ biology?).

[R package] I've tested the MICSQTL package and it worked fine. However, it would be helpful if the vignette walked the users through a working example, not stating "This code chunk assumes that you already have a bulk protein matrix named protein_data and annotation information for proteins called anno_protein". Other R packages often allow to import the prepared example (data(dataset)) and that's helpful for users as they can explore the exact format of input - which is oftentimes easier than reading a documentation. Nevertheless, I uploaded my own data, and the software seemed to work fine.

Minor:

- Line 171: Isn't there a typo in reference to the figure? I think you want to refer to 3A, not 3D (please double check all references).
- Line 192: I think you want to refer to S5, not S4 (please double check all references).
- Line 60: excitatory -> excitatory

Reviewer #2 (Remarks to the Author):

Deconvolution is a powerful method for identifying cell-type-specific transcriptomic signals without the need for cellular segmentation. However, existing methods in this category typically require a cell-type-specific expression signature matrix and have not been extended to proteomics research. The authors introduce a novel algorithm and tool designed to dissect bulk proteomes by leveraging shared information between transcriptome and proteome data. They first identify potential cell marker proteins by integrating RNA and protein bulk expression profiles. Subsequently, it jointly quantifies cell abundance in mixed proteomes without relying on a reference signature matrix. This approach enables downstream analyses, such as cell-type-specific protein Quantitative Trait Loci (cspQTL) mapping. However, the authors need to address the following issues:

1. The authors have introduced a novel algorithm for proteomics deconvolution; however, it appears that there is no comparison with other methods. Given the emergence of several new methods like scpDeconv, we recommend that the authors conduct a comprehensive comparison by evaluating a broader range of methods for this task.
2. The objective of proteomics deconvolution by MICSQTL is not clear. It would be helpful to understand whether proteomics deconvolution can potentially offer insights into biological significance.
3. It would be advantageous if the proposed MICSQTL method could perform proteomics deconvolution across different tissues or datasets. We recommend that the authors investigate whether MICSQTL can address diverse types of proteomics.
4. The performance of the proposed MICSQTL method may be influenced by hyperparameters. Therefore, it is advisable for the authors to assess how their model's performance is impacted by various hyperparameters and provide insights into their process for selecting the optimal hyperparameters.

Reviewer #3 (Remarks to the Author):

The proposed deconvolution provide modest improvement over other methods. Im supportive of the manuscript but it is a bit brief so be fully convinced. The associated R-package is a real strength

Major comments:

The case study is good but it would be nice to another application. It would also be good to see another proteomics deconvolution method applied side-by-side here to see if the same conclusions where

reached with a competing method.

The narrative is a little bit confusing jumping between MS based proteomics and CITE-seq. If I understand correctly there was never actually validation on a MS dataset?

Can the method handle a random-effect, where the deconvolution depends on some additional heterogeneity.

The authors present joint non-negative matrix factorisation as if it is novel but many others have already employed this approach for other problems.

Minor comments:

Figures are hard to read and some are missing labels.

Fig.2B. I think the axis should be scaled evenly.

I don't think $r > 0.3$ is a strong correlation

RE: COMMSBIO-23-3899-T

Title: Multimodal joint deconvolution and integrative signature selection in proteomics

Reviewer #1:

1. *The Pan et al introduces a novel algorithm, MICSQTL, designed for dissecting bulk proteomes by leveraging information shared between transcriptome and proteome data. The method identifies potential cell marker proteins and quantifies cell abundance in mixture proteomes, enabling downstream analyses such as cell-type-specific protein Quantitative Trait Loci (cspQTL) mapping. The authors validate the algorithm using various datasets, including pseudo bulk expression profiles from CITE-seq, synthetic bulk multimodal expression, and real human brain transcriptome-proteome. The results demonstrate the validity and power of MICSQTL, particularly in accurately estimating cell abundances in proteins. The tool is applied to study bipolar disorder and schizophrenia, showcasing its potential to reveal disparities in astrocyte and oligodendrocyte abundances between cases and controls. The tool yields more accurate cell proportion estimates in the prefrontal cortex compared to Cibersort (referred to as CBS), effectively representing the actual distribution, such as accurately depicting the prevalence of oligodendrocytes and maintaining a 1:3 ratio between Inhibitory (In) and Excitatory (Ex) neurons. Nevertheless, I ask some questions about these results later - as well as for cspQTL result that was also used to demonstrate the utility of the tool.*

Response: Thank you for the thorough and positive summary of our manuscript.

2. *The authors report disparities in astrocyte and oligodendrocyte abundances between cases and controls. However, there is a limited discussion about the direction of that difference (higher in controls or vice versa?) and how it matches the existing findings in terms of the directionality of the change. It is not clear whether the input matrix was somehow normalized / residualized, i.e. whether the effect of technical covariates was regressed out. Are those reported differences statistically significant? Is there any other cohort that could be used to replicate these findings? I am asking also before I previously saw papers reporting the changes mostly (but not only) in glutamatergic neurons.*

Response: We thank the reviewer for inquiring about the disparities between cases and controls. The real data of schizophrenia (SCZ) and bipolar disorder (BP) are profiled in a multi-center cohort study with $n = 45$ SCZ and $n = 25$ BP cases and $n = 194$ healthy controls. These samples were collected from two clinical centers with strong confounding factors such as age, gender, and race, although the technical (bioinformatics) covariates were normalized in the preprocessing of multi-omics data, as

described in the Supplementary File.

Given the unbalanced sample size and the confounding clinical covariates, our current analysis focused on the validity of deconvoluted proportions in the context of **population-wide cellular composition** regardless of disease phenotype rather than the case-to-control disparities. **Figure 4 (A-B) and Supplementary Table S1** show that JNMF successfully improves (or corrects) the fractions of inhibitory neurons and microglia estimated by CIBERSORT on transcriptomes. We performed Wilcoxon tests on the JNMF cell proportions for each pair of subgroups, showing microglia and oligodendrocytes significantly differ (adjusted p-value < 0.05) between SCZ vs young controls (under 70 years of age). However, this result may be biased due to the limitations in the study design and cannot be validated by an existing scRNA-seq study in SCZ. Two recent large-cohort studies in SCZ [1, 2] using scRNA-seq profiling for human brain tissues did not find significant cellular disparities between SCZ and controls, possibly because of the noises introduced by cell clustering, labeling, and limited cell counts in rare cellular subpopulations. Thus, we did not emphasize the cell abundance difference between cases and controls in the current work. On **page 11, lines 209 to 213**, we explained that the potential case-to-control disparities in glia cells need further single-cell experiments or an external large cohort for validation.

3. *Would it be possible to generate sample-level estimates as well as per-cell-type exact outputs (p-values & adj. p-values) as supplementary tables? The results should be better described quantitatively (e.g. as Supplementary Table) as well as in the biological context (are those implicated genes established in SCZ biology?).*

Response: Thank you for the great suggestion to report sample-level estimates as Supplementary Tables and describe the results in the biological context of SCZ. **Supplementary Tables S1 and S3** show sample-wise cellular proportions in the proteomes of the human prefrontal cortex and breast cancer, while the adjusted p-values for cspQTL analysis are listed in Supplementary Table S2. In the revised manuscript, we also reported the significant cspQTL results in the context of neurodevelopmental and psychiatric diseases; please refer to **page 13, lines 225 to 236**.

4. *[R package] I've tested the MICSQTL package and it worked fine. However, it would be helpful if the vignette walked the users through a working example, not stating "This code chunk assumes that you already have a bulk protein matrix named protein_data and annotation information for proteins called anno_protein". Other R packages often allow to import the prepared example (data(dataset)) and that's helpful for users as they can explore the exact format of input - which is oftentimes easier than reading a documentation. Nevertheless, I uploaded my own data, and the software seemed to work fine.*

Response: We thank the reviewer for testing our Bioconductor package with their own data. We have revised the vignettes on Github and Bioconductor to clarify that users can import the prepared example data using `data(dataset)`.

5. *Minor: Line 171: Isn't there a typo in reference to the figure? I think you want to refer to 3A, not 3D (please double check all references). Line 192: I think you want to refer to S5, not S4 (please double check all references). Line 60: exitatory - > excitatory*

Response: We apologize for the typos and reference. In the revised manuscript, we have corrected typos and reference to figures.

Reviewer #2:

1. *Deconvolution is a powerful method for identifying cell-type-specific transcriptomic signals without the need for cellular segmentation. However, existing methods in this category typically require a cell-type-specific expression signature matrix and have not been extended to proteomics research. The authors introduce a novel algorithm and tool designed to dissect bulk proteomes by leveraging shared information between transcriptome and proteome data. they first identify potential cell marker pro-*

teins by integrating RNA and protein bulk expression profiles. Subsequently, it jointly quantifies cell abundance in mixed proteomes without relying on a reference signature matrix. This approach enables downstream analyses, such as cell-type-specific protein Quantitative Trait Loci (cspQTL) mapping. However, the authors need to address the following issues:

Response: Thank you for the thorough and positive comments on our work.

2. *The authors have introduced a novel algorithm for proteomics deconvolution; however, it appears that there is no comparison with other methods. Given the emergence of several new methods like scpDeconv, we recommend that the authors conduct a comprehensive comparison by evaluating a broader range of methods for this task.*

Response: We appreciate the suggestion to add another proteomics deconvolution method. We aim to demonstrate the performance of our algorithm JNMF without using reference single-cell (type) proteomics data. Hence, most traditional reference-based deconvolution algorithms cannot be fairly compared. The tool scpDeconv allows deconvolution of bulk proteomes by training the deep learning model on scRNA-seq reference data, although the performance was less ideal compared to using single-cell proteomics data for training. Therefore, in the revised manuscript, we added scpDeconv to the simulation study and the multi-omics data of breast cancer as a competing method. We did not apply scpDeconv in the deconvolution of CITE-seq pseudo-bulk proteomics data because this type of benchmark for scpDeconv was performed and published (see **Figure 2 (C-D)** in [3]) using a reference single-cell transcriptome or proteome for training. The results on **page 10, lines 183 to 191**, and **page 15, lines 263 to 274**, show that JNMF consistently outperforms scpDeconv when high-quality single-cell proteomics reference data are lacking.

3. *The objective of proteomics deconvolution by MICSQTL is not clear. It would be helpful to understand whether proteomics deconvolution can potentially offer insights into biological significance.*

Response: Thank you for the great suggestion regarding the goal of proteomics deconvolution. Proteomics deconvolution provides quantification of cell abundance and cell-type-specific expression of proteins not detectable in single-cell proteomics technologies. In the Introduction section, we explained the biological significance of proteomics deconvolution on **page 3, lines 47 to 50, and lines 59 to 62**.

4. *It would be advantageous if the proposed MICSQTL method could perform proteomics deconvolution across different tissues or datasets. We recommend that the authors investigate whether MICSQTL can address diverse types of proteomics.*

Response: We agree with the reviewer that it's important to benchmark the performance of MICSQTL across different tissues or datasets. We used two different datasets with ground truth of sample-wise cell proportions to benchmark the outperformance of our method: the CITE-seq pseudo-bulk data from human PBMC samples and the synthetic mass spectrometry (MS) proteomics data mimicking human brain tissues. In the revised manuscript, we also added another multi-omics dataset from breast cancer (BC) tumor tissues and applied competing methods (CBS, JNMF, scpDeconv) to both real datasets. The results in CITE-seq pseudo-bulk data (**Figure 2**), the synthetic and real MS proteomics data from human brain tissues (**Figures 3&4**), and MS proteomics data from BC tumors (**Figure 5**) all demonstrated robust and accurate cellular compositions predicted by our method JNMF.

5. *The performance of the proposed MICSQTL method may be influenced by hyperparameters. Therefore, it is advisable for the authors to assess how their model's performance is impacted by various hyperparameters and provide insights into their process for selecting the optimal hyperparameters.*

Response: Thank you for this insightful comment. We agree that the hyperparameters in this model have a tremendous impact on performance. Therefore, we performed extensive experiments on the hyperparameters, i.e., initial values and step size in Projected Gradient Descent (PGD). The results showed that robust and superior performance is achieved by initializing the multi-modal pure cell expressions X_1 , X_2 and the shared cell counts fractions p through an external RNA-seq signature

matrix and the RNA proportions pre-estimated by CBS on this signature matrix. See the Method section for initialization details. The output cellular compositions with this initialization approach are less sensitive to the PGD step size compared to the previous version of JNMF.

In the revised simulation study, we used different values of the PGD step size in CITE-seq pseudo-bulk data and illustrated that adjustment in step size did not lead to substantial changes in performance (**Figure 2, Supplementary Figures S1-S5**). Furthermore, we randomly generated $n = 50$ epi-cates of the signature matrix with small vs. large noises to run CBS on RNA-seq and JNMF on bulk multi-omics data. The results in **Figure 3** demonstrate that JNMF significantly outperformed CBS, TCA, and scpDeconv across different scenarios and signature matrices. In the R package, we provided a minimum PGD step size as default, which is a value used in the simulation study. Users can adjust the scaling of input bulk multi-omics data by log or MinMax along with the PGD step size to achieve a relatively ideal deconvolution result for downstream analysis.

Reviewer #3:

1. *The proposed deconvolution provides modest improvement over other methods. Im supportive of the manuscript but it is a bit brief so be fully convinced. The associated R-package is a real strength.*

Response: Thank you for the positive comment about our manuscript.

2. *The case study is good but it would be nice to another application. It would also be good to see another proteomics deconvolution method applied side-by-side here to see if the same conclusions were reached with a competing method.*

Response: We appreciate the suggestion to add another proteomics deconvolution method. In the revised manuscript, we included scpDeconv as a competing method in the simulation study and applied it to the bulk multi-omics data of breast cancer. However, for the real MS proteomics data of the human brain prefrontal cortex, we were unable to apply scpDeconv to the single-cell proteomics reference data due to unavailability. Therefore, we utilized scpDeconv, CIBERSORT, and our method JNMF on the bulk MS proteomes in a breast cancer study for a side-by-side comparison. In this comparison, a single-cell CyTOF proteomics dataset was used as a reference to train scpDeconv. The results are described on **page 15, lines 259 to 274**, and **Figure 5**. Overall, scpDeconv, based on single-cell proteomics reference data measured by CyTOF, yielded slightly better performance than JNMF-AJ-RF. However, the results of scpDeconv were not stable and became less ideal if users allowed the imputation of additional cell marker proteins in the single-cell reference during training.

3. *The narrative is a little bit confusing jumping between MS based proteomics and CITE-seq. If I understand correctly there was never actually validation on a MS dataset?*

Response: We apologize for the confusion between MS-based proteomics and CITE-seq data. We used CITE-seq pseudo-bulk multimodal data from PBMC to demonstrate the validity and accuracy of our algorithm since the cells in this dataset were well annotated by weighted-nearest neighbor analysis in Seurat. Another reason for the lack of scMS proteomics data in validation is the rarity of public sources of scRNA-seq and scMS measurements matched by tissue. The PBMC CITE-seq data seems to be the most viable public resource for generating tissue-matched high-throughput pseudo-bulk multi-omics data from ground truth.

4. *Can the method handle a random-effect, where the deconvolution depends on some additional heterogeneity?*

Response: Instead, our algorithm iteratively updates the cell type proportions and individualized multimodal reference profiles by adapting the parameter optimization to each tissue, thus accounting for between-subject heterogeneity. For further details on the statistical model employed in our method, please refer to **page 18, lines 328 to 338**, and **page 19, lines 339 to 344**. To clarify the statistical model used in our method, we renamed our deconvolution algorithm as JNMF throughout

the manuscript.

5. *The authors present joint non-negative matrix factorization as if it is novel but many others have already employed this approach for other problems.*

Response: We thank the reviewer for pointing out the utilization of JNMF in other problems. We agree that this algorithm has been employed in other areas, but this approach has not been adopted in cell type deconvolution or implemented in an R package. The aim of using JNMF in deconvolution is to accurately estimate the cell count fractions shared between multimodal data. This is a novel framework describing multi-modal cellular compositions, which has not been considered in existing deconvolution methods. Recent publications utilized a Bayesian algorithm [4, 5] to perform multi-omics deconvolution, but their model did not incorporate the latent cell counts shared between transcriptome and proteome.

6. *Figures are hard to read and some are missing labels. Fig.2B. I think the axis should be scaled evenly. I don't think $r > 0.3$ is a strong correlation.*

Response: Thank you for pointing out the figure format issue and the correlation coefficients. We updated **Figure 2** to show updated CITE-seq results and added **Figure 3** to show simulation results. In the CITE-seq pseudo bulk data, we used z-scores to scale the purified protein and RNA expression, with axes rescaled evenly (see **Figure 2 (C-H)** and **Supplementary Figures S1-S5**). The improved JNMF algorithm yielded better Pearson correlations between the true and predicted values for either cellular fractions ($r > 0.9$) or cell-type-specific purified expressions of protein ($r > 0.6$) and RNA ($r > 0.9$).

References

- [1] Ruzicka, B. *et al.* Single-cell dissection of schizophrenia reveals neurodevelopmental-synaptic link and transcriptional resilience associated cellular state. *Biological psychiatry* **89**, S106 (2021).
- [2] Ruzicka, W. B. *et al.* Single-cell multi-cohort dissection of the schizophrenia transcriptome. *medRxiv* 2022-08 (2022).
- [3] Wang, F. *et al.* Deep domain adversarial neural network for the deconvolution of cell type mixtures in tissue proteome profiling. *Nature Machine Intelligence* **5**, 1236–1249 (2023).
- [4] Petralia, F. *et al.* Bayesdebulk: a flexible bayesian algorithm for the deconvolution of bulk tumor data. *bioRxiv* 2021-06 (2021).
- [5] Petralia, F. *et al.* Pan-cancer proteogenomics characterization of tumor immunity. *Cell* (2024).

REVIEWERS' COMMENTS:

Reviewer #1 (Remarks to the Author):

I am satisfied with the provided responses and I have no further comments except for a minor request for specifying (in the Data Availability statement) where the SCZ/BD dataset from SMRI/BSHRI can be found. If it is already there, please ignore this - but it seemed to me that it was missing, i.e. only data from other sections were referred.

Reviewer #2 (Remarks to the Author):

recommend to accept

Reviewer #3 (Remarks to the Author):

The authors have addressed my concerns.